# Effect of Irrigation with Activated Water on Root Morphology of Hydroponic Rice and Wheat Seedlings

**Xueting Yang** [1,2]**, Jun Fan** [1,3,*]**, Jiamin Ge** [1,2] **and Zhanbin Luo** [3]

1   State Key Laboratory of Soil Erosion and Dryland Farming on the Loess Plateau, Institute of Soil and Water Conservation, Chinese Academy of Sciences and Ministry of Water Resources, Xianyang 712100, China; yangxueting17@foxmail.com (X.Y.); gejiamin128@126.com (J.G.)
2   University of Chinese Academy of Sciences, Beijing 100049, China
3   Institute of Soil and Water Conservation, Northwest A&F University, Xianyang 712100, China; zbluo@nwafu.edu.cn
*   Correspondence: fanjun@ms.iswc.ac.cn

**Abstract:** Raising yields and agricultural production efficiency is the primary goal of realizing modern agriculture. As a low-cost and environmentally friendly technology, activated water has the potential to increase crop yields and improve water and fertilizer use efficiency, but much research is still needed to make this technology widely available in the field. Hydroponic experiments were conducted to investigate the effects of magnetized water, aerated water, and magnetized aerated water on rice and wheat seedling and root growth. The results showed that aerated water irrigation significantly increased plant height by 5.1–9.6%, leaf area by 21.1%, and aboveground biomass by 14.8–16.3%, respectively. Aerated water irrigation also significantly promoted rice root biomass, maximum root length, total root length, total root surface area, and especially the proportion of roots less than 0.5 mm in diameter, indicating that aerated water enhances the growth of rice seedlings mainly by promoting root growth, especially fine root (D ≤ 0.5 mm) growth. The maximum root length and total root volume of wheat roots under magnetized water irrigation treatment were increased by 7.7–8.6% and 17.2%, respectively, resulting in a significant increase in aboveground dry biomass by 13.6%. Magnetized water and magnetized aerated water irrigation also promoted the growth of rice seedlings and roots. In contrast, aerated water and magnetized aerated water irrigation exhibited an inhibitory effect on the growth of wheat seedlings and roots. Therefore, activated water has different effects on different crops in hydroponics, and more research is needed in the future to determine the conditions for the application of activated water in agriculture.

**Keywords:** magnetized water; aerated water; magnetized aerated water; root length

## 1. Introduction

Food resources are the basis of human survival and development. During the last few decades, crop yields have improved significantly due to improvements in technology and agricultural practices [1], the development of new cultivars [2], the utilization of chemical fertilizers [3], and the realization of mechanization [4]. However, the demand for food resources continues to increase as the world population grows, and it is reported that the global population will reach 9.1 billion by 2050 [5]. At the same time, the increase in extreme weather events in recent years has posed an enormous challenge to food production [6]. Therefore, increasing the yield of food while reducing environmental impacts (e.g., greenhouse gas emissions, biodiversity loss, land-use change, and ecosystem service loss) has become an urgent matter to address. Due to the continuous reduction of arable land in China, improving yields and agricultural production efficiency is an effective way to increase food production [7], which is also critical to achieving agricultural modernization [8]. Among the many methods to improve yields and agricultural production efficiency,

activated water technology has received extensive attention due to its simple operation, low cost, and environmental friendliness [9].

Activated water obtained by treating water with physical techniques (e.g., through oxygenation techniques, increasing the dissolved oxygen content in water, or passing water through magnetic fields of specific field strength) has been widely used in agricultural production since the first part of the 19th century [9–12]. Many studies have shown that irrigating crops with activated water can significantly increase crop yields and improve fruit quality [13–17]. After the activation of water, the structure and physical properties have been changed, the average distance between water molecules increases, some hydrogen bonds become weaker or even broken, the number of free monomeric water molecules and dimeric water molecules in water increases, and dissolved oxygen content increases [9,11,18,19]. Otsuka and Ozeki [20] found that the contact angle of distilled water was significantly reduced from 65° to 57.5° after magnetization. The results of Esmaeilnezhad et al. [18] showed that the surface tension of water decreased, and the pH increased after magnetization of both pure and saline water. Zhao et al. [21] concluded that magnetization and oxidation of water increased pH and maintained it for about 60 h, reduced surface tension and viscosity coefficient by 7.4–15.4%, maintained it for about 8–10 h, increased dissolved oxygen concentration, and detected the formation of hydroxyl radicals. These changes in water properties help improve the soil environment after the treated water enters the soil, increase soil infiltration capacity, enhance soil water content, raise soil dissolved oxygen concentration, improve root zone aeration, accelerate soil organic matter decomposition, and promote water and nutrient uptake and utilization by the root system [22–25]. At the same time, treating seeds with activated water can enhance the activity of major enzymes in the seeds, improve seed vigor, and promote seed germination [26–29]. In addition, activated water irrigation increases the photosynthetic rate, transpiration rate, chlorophyll content, leaf area of the leaves and improves soil water and fertilizer utilization [15,30,31].

However, some studies have also found that activated water has a negative or uncertain effect on the growth of crops because the properties of activated water are affected by the activation time, activation equipment, water quality, and other factors [32–36]. Ahmed et al. [29] studied the influences of four categories of nanobubbles water (air, oxygen, nitrogen, and carbon dioxide) on seed germination and seedling growth of different crops. They found that air nanobubbles and carbon dioxide nanobubbles did not significantly promote carrot and soybean seeds´ germination compared to tap water. Air nanobubbles negatively affected tomato growth, reducing the number, length, and width of leaves. Almaghrabi and Elbeshehy [37] treated nine wheat varieties with magnetized water and found that magnetized water treatment significantly decreased the germination rate, seedling growth parameters, and protein bands numbers of cultivar Sakha93 and Masr1. At the same time, no significant difference was observed for Masr1cv compared with control. Maheshwari and Grewal [38] explored the impacts of magnetically treated tap water, recycled water, and saline water on yield and water use efficiency of snow peas, celery, and peas. The results showed that all types of magnetically treated water had no significant effect on the yield and water productivity of peas and had no significant effect on the total water use of the three vegetables. Although the effects of activated water on crop growth and yield, especially magnetized water, are still controversial, it warrants continued scientific research because of the cheap and environmentally friendly features of activated water technology. Moreover, we found few studies combining two different types of activated water to study their superimposed effects. The mechanism of crop growth affected by activated water is not fully clear. Therefore, much research on the application of activated water in agriculture is still needed to clarify the qualifying conditions and mechanisms of activated water for crop yield increase to apply this cheap measure widely.

In this study, we used hydroponics to investigate the effects of magnetized and aerated water on the growth of rice (*Oryza sativa* L.) and wheat (*Triticum aestivum* L.), and explored whether combining magnetized water with aerated water (magnetized aerated

water) showed a superimposed effect on crop growth. The study´s objectives were to (1) explore the effect of different types of activated water applied on rice and wheat seedlings, (2) investigate the effect of different types of activated water on the root characteristics of rice and wheat in an attempt to explain why activated water affects seedling growth.

## 2. Materials and Methods

### 2.1. Experimental Site

Two hydroponic experiments (for rice and wheat) were conducted from July to September 2018 in the greenhouse of Northwest A&F University. The natural light was used in the greenhouse, the temperature was 10–30 °C, and the relative humidity was between 45–75%. The greenhouse included supplementary light, ventilation, external shading, internal insulation, wet curtain-fan cooling, and winter heating systems.

### 2.2. Experimental Design

With distilled water (CK) as the control, three types of activated water, including magnetized distilled water (magnetized water, MW), aerated distilled water (aerated water, AW), and magnetized aerated distilled water (magnetized aerated water, MAW), were applied to rice and wheat seedlings, with a total of 4 treatments for each crop. Four replications of each treatment. Rice seeds (Xiuzhan 15) were surface-sterilized by 10% $H_2O_2$ for 30 min, rinsed with distilled water three times, and soaked in distilled water for two days. The seeds were then germinated on a moistened filter paper in a tray at 28°C for 15 days. Wheat seeds (Xiaoyan 22) were surface-sterilized by 10% $H_2O_2$ for 10 min, washed with distilled water three times, and soaked in distilled water for eight hours. The seeds were then germinated on a moistened filter paper in a tray at 25 °C for eight days. After rice and wheat germination, uniform rice and wheat seedlings of similar size and vigor with three leaves were transplanted to the hydroponic container (30 cm × 20 cm × 14.7 cm) containing 3 L of nutrient solutions and fixed on a foam board with a sponge. Each hydroponic container accommodated four plants, i.e., each treatment included 16 plants. The nutrient solution was cultivated with different treated water and half-strength Hoagland solution, and the nutrient solution was refreshed every three days, with the pH maintained at 6.0. Distilled water was used in this study to avoid the effect of ions contained in tap water on the hydroponic experiment.

### 2.3. Preparation of Activated Water

MW was prepared by circulating magnetized 15 L distilled water through the magnetic treatment device for 20 min. The magnetic treatment device consists of a water tank, peristaltic pump, pipe, and permanent magnet with a field intensity of 3000 GS. The permanent magnet was installed on the outside of the pipe, and the distilled water was magnetized as it passed through the pipe. AW was prepared by circulating oxygenation 15 L distilled water through the aeration system for 20 min. The aeration system consists of a micro-nano bubble generator (Shanghai ZhongJing Environmental Protection Technology Company Limited, Shanghai, China) and a water tank. MAW was prepared by circulating oxygenation 15 L MW through the aeration system for 20 min. The principles of MW and AW production are the same as those of Zhao et al. [21] and Zhu et al. [19] respectively, so their conclusions regarding the physicochemical properties of water are applicable to the present study.

### 2.4. Sampling and Measurements

Plant height and the maximum root length of the rice and wheat were measured every week during the whole growth process, and then the growth rate of plant height and maximum root length were calculated. The plant height and the maximum root length of each plant were measured three times and the mean from one plant was then counted as one biological replicate. In total, there 16 biological replicates for each graph. At 40 days from transplantation, rice and wheat plants were harvested and divided into roots and above-

ground parts. The fresh weights of the aboveground and root parts, leaf number and leaf area of rice and wheat, tiller number of wheat, and stem diameter of rice were determined for each plant. The leaf area of rice and wheat was calculated by leaf length × leaf width × 0.75. These data were measured once per plant, and the average of 16 plants was used for the graph. All roots of one plant were rinsed with distilled water and then placed on a plexiglass container filled with a little water and carefully spread out to avoid overlap or crossover between the roots. Images of the roots were taken with an Epson scanner (Epson Perfection V700 Photo, Beijing, China). WinRHIZO software was used to determine the root length, root surface area, root volume, and mean root diameter from the images. The roots of each plant were scanned once, and eight plants were randomly selected for scanning under one treatment. The results obtained were the average of the eight plants. The aboveground and roots were oven-dried at 65 °C for 48 h to measure the aboveground and root dry weights of each plant. The mean of a total of 16 plants per treatment was taken for graphing. The root-shoot ratio was calculated from the dry weights.

### 2.5. Statistical Analysis

Data were recorded in Excel 2016. Statistical analysis was conducted using SPSS 22.0 software. One-way and two-way analysis of variance, followed by Duncan's multiple comparison test, was performed to find the differences between treatments.

## 3. Results

### 3.1. Plant Height

The response of rice and wheat height to activated water irrigation was presented in Figure 1. Significant differences were observed among different treatments on the 19th day after rice plant transplanting but on the 40th day after wheat plant transplanting. Compared with CK, irrigation with MW, AW and MAW increased the rice plant height by 2.2–8.0%, 5.1–9.6% and 1.0–6.1%, respectively. The difference between AW and CK showed statistical significance at 19, 26 and 40 days after rice transplanting. In the whole growth stage of rice, the plant height growth rate of MW, AW, and MAW were 0.86, 0.92 and 0.84 cm d$^{-1}$, respectively, which increased by 6.3%, 13.7% and 4.5% compared with the control. However, for wheat, compared with CK, irrigation with MW increased by 1.0–2.3%, while AW and MAW decreased by 1.0–4.3% and 0.3–2.0%, respectively, which it is not statistically significant. Briefly, activated water had a significant effect on the height of rice seedlings, while it had almost no effect on the height of wheat seedlings.

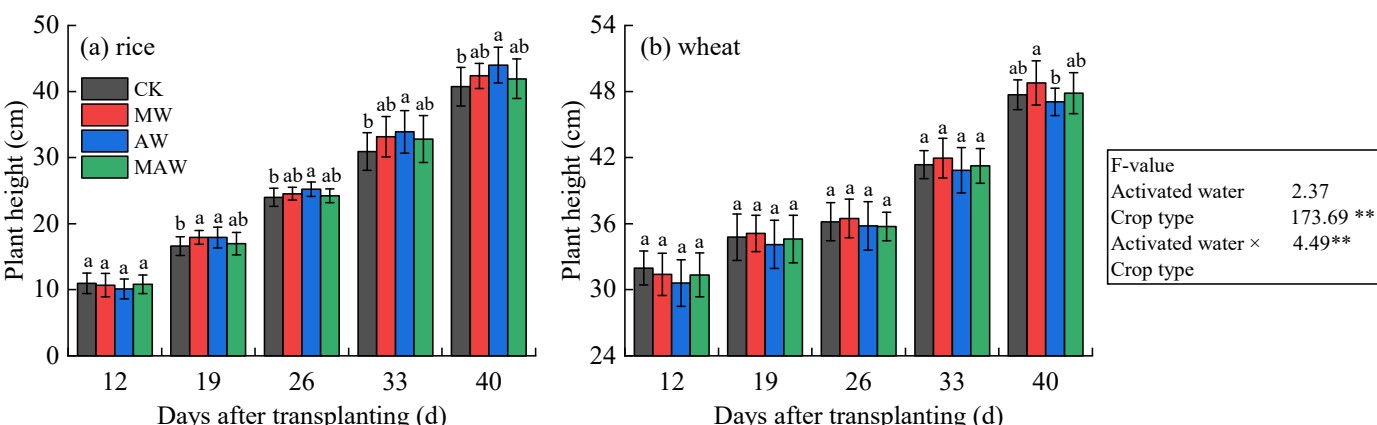

**Figure 1.** Changes in plant height of rice and wheat in the hydroponic system under active water irrigation at different growth periods. CK, distilled water; MW, magnetized water; AW, aerated water; MAW, magnetized aerated water. Different letters indicate significant differences among treatments at $p < 0.05$. The error bars indicate standard deviations. ** significant at $p < 0.01$.

### 3.2. Growth Characteristics

The growth parameters for rice and wheat are shown in Table 1 were obtained at harvest time. In comparison with CK, irrigation with AW significantly increased the fresh and dry weight of the aboveground and the root dry weight for rice by 16.3%, 14.8% and 22.4%, respectively. The aboveground biomass (15.7% and 18.3%) of rice treated with MW irrigation was also significantly higher than CK. Compared with CK, irrigation with AW and MW also significantly increased the rice leaf area by 21.1% and 15.7%, respectively. Irrigation with MAW significantly reduced the number of leaves and stem diameter of rice compared to CK.

Different from the results of rice, the effects of activated water irrigation on wheat growth parameters did not show significant differences, except for aboveground dry biomass. The effects of the treatments on the leaf area and aboveground fresh and dry biomass of wheat were such that MW > CK > MAW > AW. This indicated that magnetized water irrigation increased wheat aboveground biomass, while aerated water and magnetized aerated water decreased it compared with the control.

### 3.3. Rice and Wheat Root System

3.3.1. Maximum Root Length

Activated water irrigation treatment had no significant effect on the growth of the maximum root length of both rice and wheat during the first 26 days after transplanting (Figure 2). At 33 and 40 days after transplanting, the maximum root length of rice treated by aerated water irrigation was significantly higher than that of other treatments (Figure 2a). MW, MAW, AW increased by 9.4%, 2.6% and 5.1% at 33 days after rice transplanting and 15.4%, 7.0% and 11.0% at 40 days, respectively. At 33 days of rice transplantation, the effects of the treatments on the maximum root length were as follows: AW > MW > MAW > CK. The irrigation treatments of MW and MAW increased by 6.6–7.8% and 4.0–4.1%, respectively, compared with CK. During the whole period, the growth rate of the maximum root length of rice under aerated water irrigation was the highest, which was 0.67 cm d$^{-1}$, 29% higher than that under distilled water irrigation. The growth rates of distilled water, magnetized water, and magnetized aerated water were 0.52, 0.60 and 0.57 cm d$^{-1}$, respectively. Compared with distilled water, magnetized water and magnetized aerated water irrigation treatment increased by 14.4% and 8.7%, respectively.

Similar to rice, the maximum root length of wheat with different treatments showed a significant difference at 33–40 days after transplanting (Figure 2b). MW had the highest maximum root length, and MAW had the lowest maximum root length after 33 days. MW was significantly higher than the control, increasing by 7.7–8.6%. Irrigation with MAW decreased by 1.2% compared with CK on the 40 days after wheat transplanting. The values of the growth rate of the maximum root length followed the order of AW > MW > MAW > CK. Compared with CK, Irrigation with AW, MW and MAW increased by 23.9%, 23.1% and 5.2%, respectively.

3.3.2. Root System Morphology

Compared with distilled water, activated water irrigation significantly promoted the growth of rice roots (Figure 3). The total root length and total root surface area under AW irrigation were 2870 cm and 177 cm$^2$, respectively, which were significantly increased by 33.8% and 27.8% compared with CK (2145 cm and 139 cm$^2$). While the total root length and total root surface area under MW and MAW irrigation were 2266 cm, 2346 cm and 155 cm$^2$, 153 cm$^2$, respectively, which increased by 5.7–11.9% than CK, with no significant difference between treatments. The total root volume treated with AW, MW, and MAW increased by 16.7%, 21.1%, and 10.3%, respectively, compared with CK. For mean root diameter, the MW irrigation treatment was 8.6% higher than the CK, while the AW and MAW were 4.8% and 1.3% lower than the CK. However, the total root volume and mean root diameter did not differ significantly between irrigation with activated water and distilled water.

**Table 1.** Effect of activated water irrigation on rice and wheat growth parameters during harvest.

| | Treatments | Leaf Number | Tiller Number | Stem Diameter (mm) | Leaf Area (cm² Plant⁻¹) | Fresh Weight of Aboveground (g Plant⁻¹) | Dry Weight of Aboveground (g Plant⁻¹) | Fresh Weight of Root (g Plant⁻¹) | Dry Weight of Root (g Plant⁻¹) | Root-Shoot Ratio |
|---|---|---|---|---|---|---|---|---|---|---|
| Rice | CK | $9.7 \pm 0.5a$ | - | $0.84 \pm 0.09a$ | $57.8 \pm 8.6c$ | $1.50 \pm 0.20b$ | $0.24 \pm 0.04b$ | $0.66 \pm 0.29a$ | $0.048 \pm 0.010b$ | $0.20 \pm 0.01ab$ |
| | MW | $9.9 \pm 0.7a$ | - | $0.83 \pm 0.09a$ | $66.8 \pm 7.1ab$ | $1.74 \pm 0.17a$ | $0.29 \pm 0.03a$ | $0.74 \pm 0.36a$ | $0.052 \pm 0.007ab$ | $0.18 \pm 0.01b$ |
| | AW | $10.0 \pm 0.0a$ | - | $0.88 \pm 0.07a$ | $69.9 \pm 8.3a$ | $1.74 \pm 0.28a$ | $0.28 \pm 0.05a$ | $0.81 \pm 0.41a$ | $0.059 \pm 0.010a$ | $0.21 \pm 0.02a$ |
| | MAW | $8.8 \pm 0.7b$ | - | $0.61 \pm 0.1b$ | $61.5 \pm 11.9bc$ | $1.65 \pm 0.30ab$ | $0.28 \pm 0.05ab$ | $0.70 \pm 0.37a$ | $0.050 \pm 0.009b$ | $0.18 \pm 0.02b$ |
| Wheat | CK | $12.5 \pm 1.2a$ | $2.7 \pm 0.7a$ | - | $144.4 \pm 22.4a$ | $4.55 \pm 0.56a$ | $0.65 \pm 0.05b$ | $3.36 \pm 0.59a$ | $0.130 \pm 0.019a$ | $0.20 \pm 0.03a$ |
| | MW | $12.8 \pm 1.4a$ | $2.8 \pm 0.7a$ | - | $152.8 \pm 17.3a$ | $4.80 \pm 0.52a$ | $0.74 \pm 0.12a$ | $3.50 \pm 0.57a$ | $0.134 \pm 0.020a$ | $0.19 \pm 0.03a$ |
| | AW | $12.2 \pm 1.6a$ | $2.7 \pm 0.7a$ | - | $137.7 \pm 14.6a$ | $4.27 \pm 0.60a$ | $0.61 \pm 0.09b$ | $3.40 \pm 0.56a$ | $0.125 \pm 0.021a$ | $0.21 \pm 0.03a$ |
| | MAW | $12.6 \pm 1.6a$ | $3.1 \pm 0.7a$ | - | $141.3 \pm 21.4a$ | $4.29 \pm 0.81a$ | $0.63 \pm 0.10b$ | $3.32 \pm 0.76a$ | $0.124 \pm 0.018a$ | $0.19 \pm 0.02a$ |
| F-value | Activated water | 0.10 | - | - | 0.76 | 1.70 | 4.86 ** | 0.32 | 0.64 | 3.04 * |
| | Crop type | 2.53 | - | - | 46.69 ** | 763.15 ** | 685.07 ** | 615.72 ** | 540.77 ** | 0.64 |
| | Activated water × Crop type | 0.60 | - | - | 1.90 | 1.82 | 3.40 * | 0.14 | 1.32 | 0.06 |

Notes: CK, distilled water; MW, magnetized water; AW, aerated water; MAW, magnetized aerated water. Values in the table are mean ± standard deviation. Different letters in the same column indicate significant differences between treatments at $p < 0.05$. '-' mean not measured. * and ** indicate significant differences at the 0.05 and 0.01 probability levels, respectively.

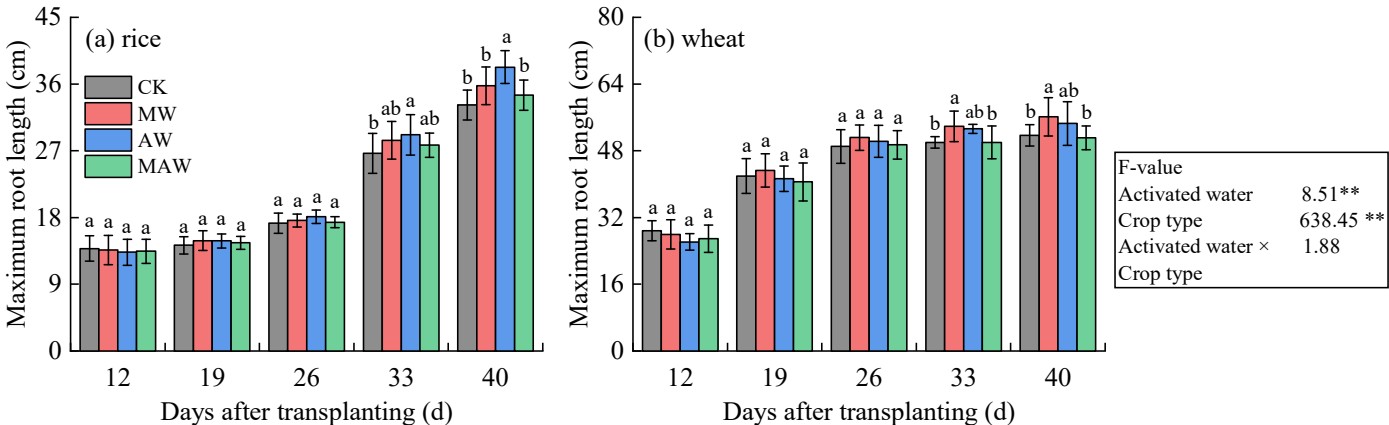

**Figure 2.** Changes in maximum root length of rice and wheat in the hydroponic system under active water irrigation at different growth periods. CK, distilled water; MW, magnetized water; AW, aerated water; MAW, magnetized aerated water. Different letters indicate significant differences among treatments at $p < 0.05$. The error bars indicate standard deviations. ** significant at $p < 0.01$.

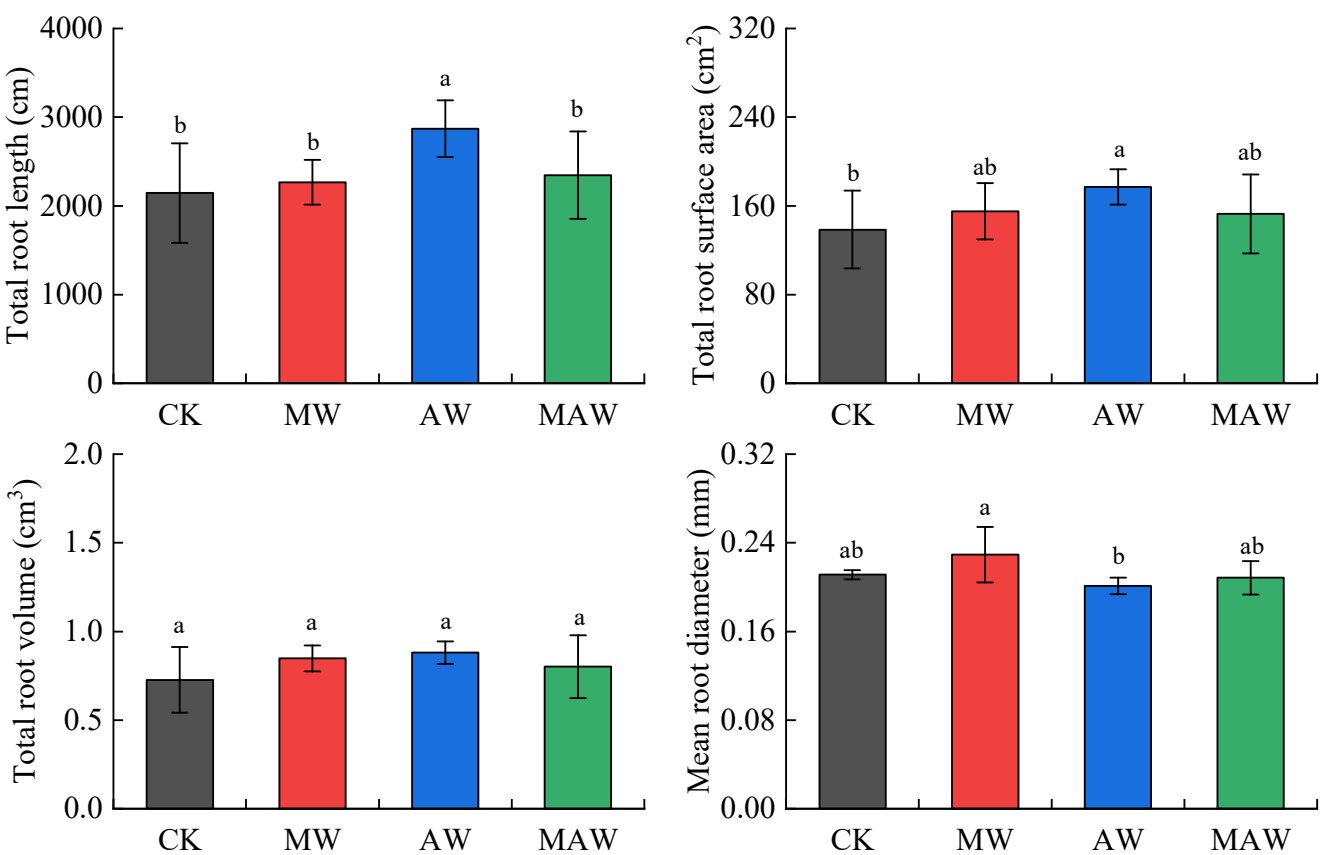

**Figure 3.** Effect of activated water irrigation on root characteristics of rice during harvest. CK, distilled water; MW, magnetized water; AW, aerated water; MAW, magnetized aerated water. Different letters indicate significant differences among treatments at $p < 0.05$. The error bars indicate standard deviations.

Unlike rice, activated water irrigation had no significant effect on total root length and total root surface area of wheat compared with distilled water (Figure 4). Compared with CK, irrigation with MW significantly increased total root volume by 17.2%, AW increased by 14.0%, and MAW decreased by 1.2%. AW irrigation significantly increased the average root diameter of wheat, while MAW decreased by 4.0% compared with the control.

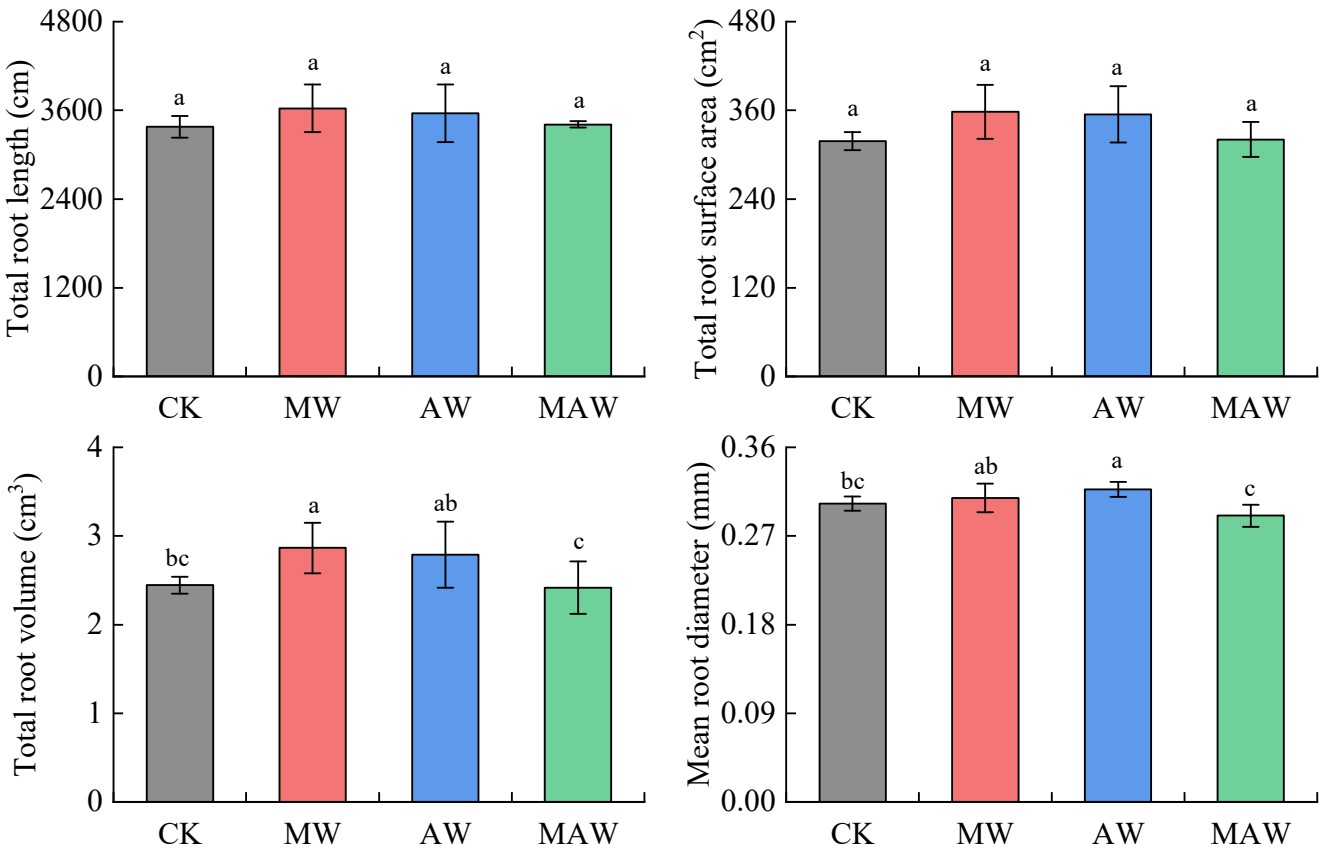

**Figure 4.** Effect of activated water irrigation on root characteristics of wheat during harvest. CK, distilled water; MW, magnetized water; AW, aerated water; MAW, magnetized aerated water. Different letters indicate significant differences among treatments at *p* < 0.05. The error bars indicate standard deviations.

### 3.3.3. Percentage of Root Length with Various Diameters to Total Root Length

The root systems of rice and wheat in diameter were mainly distributed from 0.0 mm to 0.5 mm, accounting for 90–94% of all the roots (Figure 5). For rice, activated water irrigation increased the root diameter between 0.0 mm and 0.5 mm, in the order of AW > MAW > MW > CK, but reduced the roots by 0.5–1.0 mm. The percentage of roots with diameters of 0.0–0.5 mm and 0.5–1.0 mm in AW was 93.0% and 6.8%, respectively, showing significant differences from CK. Magnetized water irrigation increased the root of wheat with a diameter between 0.0 and 0.5 mm, while aerated water and magnetized aerated water irrigation decreased it. These results indicated that activated water irrigation affected root growth mainly by increasing fine root (D ≤ 0.5 mm) content, especially for the aerated water irrigation treatment.

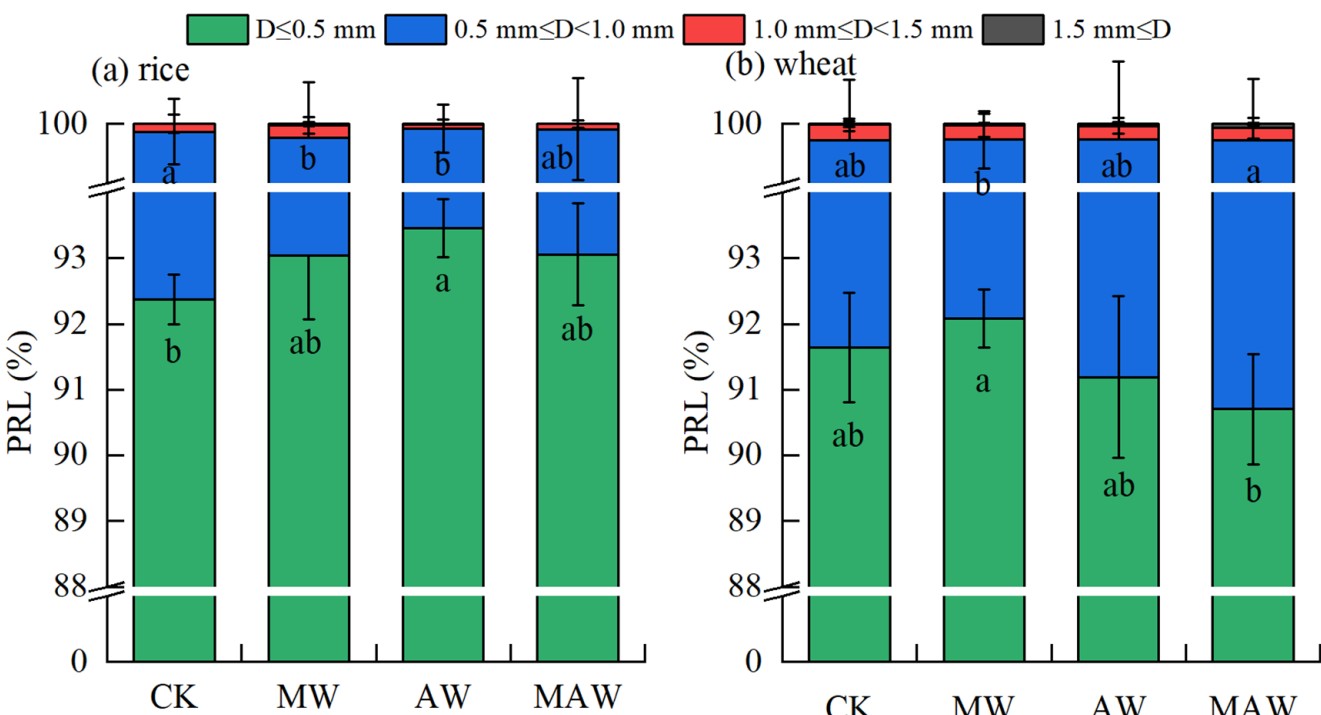

**Figure 5.** The root diameter of rice and wheat seedlings under activated water irrigation. PRL, Percentage of root length with various diameters to total root length; CK, distilled water; MW, magnetized water; AW, aerated water; MAW, magnetized aerated water. Different letters indicate significant differences among treatments at $p < 0.05$. The error bars indicate standard deviations.

## 4. Discussion

### 4.1. Effects of Irrigation with Activated Water on the Growth Characteristics of Crop

We found that plant height, leaf area, and aboveground biomass of rice under aerated water irrigation were significantly higher than those of CK, indicating that irrigation with aerated water could promote the growth of rice seedlings. Similar results were obtained by Liu and Wang [39], who used a gas supply system to supply oxygen to the root zone and found that oxygen supply had a significant effect on the growth of rice seedlings. This is mainly because the high oxygen content environment in the root zone promotes the vitality and reproduction of aerobic microorganisms, which rapidly decompose soil organic matter and accelerate the mineralization rate, thus facilitating the uptake and utilization of crops [39]. On the other hand, oxygenated irrigation increased the chlorophyll content of leaves. It enhanced the ability to absorb and convert light energy per unit leaf area, increasing the photosynthetic rate and promoting dry matter accumulation in leaves [19]. Several studies have applied aerated water by micro and nano bubble generators to greenhouse lettuce [15], tomato [40], and cucumber [41], and others have obtained aerated water by combining subsurface drip irrigation systems with air injection systems to irrigate field corn [42]. All these results show that aerated water irrigation can promote crop growth and improve crop yield. Moreover, Pendergast et al. [43] applied aerated water to drip irrigation on broadacre cotton for seven growing seasons (2005–2006 to 2012–2013) and found that the seven-year average yield was significantly higher by 10% compared to the control. Dhungel et al. [44] used Mazzei air injectors to obtain aerated water and irrigated field pineapples for 5 consecutive years (2007–2011). They arrived at a total industrial fruit yield of 73.3 t ha$^{-1}$ for the aerated treatment, which was 6% higher than the control. This is because oxygen is critical to the plant, and it transports nutrients to the cell walls and roots [40]. Root hypoxia inhibits root respiration and causes the accumulation of anaerobic microbial activity products in crop roots, which affects plant root metabolism

and leads to a rapid increase in phytohormones such as abscisic acid and ethylene, closure of leaf stomata, and a decrease in net photosynthetic rate, which seriously affects plant growth [39,45]. In addition, the aerated water in this study was aerated by a micro-nano bubble generator, which produces tiny and uniform micro-nano bubbles with low buoyancy, large relative surface area [40]. Therefore, it has the characteristics of slow rising velocity, long life span, and high internal gas density [40]. It can sustainably supply oxygen to the surrounding water and effectively avoid the 'chimney effect' caused by directly ventilating into the soil [41]. Hence, micro-nano bubble water oxygenation is considered the most efficient aerobic method [14]. A study showed that micro-nano bubble water could increase dissolved oxygen concentration in water to 36.9 mg $L^{-1}$, 4–6 times that of ordinary water, and the slight decrease takes at least 6 h to recover to the initial value [46]. The size of micro and nanobubbles is usually below 50 μm, which may be another reason to promote water nutrient uptake by crops, as the bubbles are easily attached to the surface of crop roots and change the adsorption characteristics of the root surface [15]. It also showed that the magnetized water and magnetized aerated water promoted the growth of rice seedlings to different degrees in the present experiment. This may be caused by the hydrogen bonds of water molecules being broken under the action of the magnetic field, the intermolecular forces of water being reduced, and the surface tension being decreased, which may increase the oxygen content of magnetized water [9], help the growth of rice roots, and promote the absorption and utilization of nutrients in rice.

In this study, the effects of aerated water and magnetized aerated water irrigation on the growth of wheat seedlings were different from those of rice, i.e., it reduced plant height, leaf area, and aboveground biomass of wheat. This may be correlated with the high oxygen demand of rice [47]. However, Zhu et al. [48] found that oxygenated brackish water significantly promoted wheat seed germination and seedling growth, which was mainly since oxygenated brackish water enhanced the osmoregulatory mechanism of wheat by regulating soil moisture, fertility, and gas conditions, thus improving the effects of salt stress on crop growth caused by brackish water irrigation. This is contrary to the present study results, and these different observations may be attributed to differences in water quality, wheat cultivars, time of aeration, and method of aeration. Lu et al. [35] achieved oxygenation by adding $H_2O_2$ and compared the seed germination and seedling growth of two wheat varieties with different concentrations of $H_2O_2$. The results showed that 50–200 μM $H_2O_2$ treatment stimulated seed germination of Ningchun but had no effect on Xihan seeds, and $H_2O_2$ treatment significantly reduced the root and stem length of seedlings of both varieties. This confirms that different wheat varieties respond differently to aerated water and that different oxygenation methods have different effects on the growth of wheat seedlings. In addition, Chen et al. [33] used two oxygation methods (Mazzei, Seair) to irrigate wheat. The oxygation treatment promoted wheat growth and increased wheat biomass, yield, and water use efficiency. Still, none of them reached a significant level compared to the control. Our results showed that magnetized water irrigation increased wheat plant height, leaf area, and aboveground dry biomass by 2.3%, 5.8%, and 13.6%, respectively. The results were consistent with those reported by Zhao et al. [49], who conducted a two-year (2018–19 to 2019–20) field experiments and found that irrigation with magnetized water can significantly promote winter-wheat growth, increasing yield by 10.0% and 11.1%, and improve water use efficiency and irrigation water use efficiency. Meanwhile, Hozayn et al. [50] also conducted a two-year (2009–2010 to 2010–2011) field trial using canola, and found that the seed yield and oil yield under magnetized water irrigation were 697.0 kg $fed^{-1}$ and 223.0 kg $fed^{-1}$, respectively, which increased by 38.7% and 58.5% compared to normal water. In this case, the magnetized water was usually obtained in the field by placing a permanent magnet on the irrigation water pipe. The possible reason for this is that when water is magnetized, the surface tension and viscosity coefficient of water is reduced, and the osmotic pressure and solubility increase [18]. It makes the large molecular group water into small molecule water, which is more easily absorbed by the crop cells, promoting plant growth [51]. Additionally, Zhang et al. [52]

showed that irrigation of cotton seeds with magnetized water significantly increased germination potential, germination rate, germination index, and vigor index, in the ranges of 22.2–26.3%, 16.8–22.4%, 22.2–27.0% and 47.4–78.0%, respectively. As a result, magnetized water irrigation can significantly enhance seed vigor, improve seed emergence and seedling uniformity, and promote seedling growth, resulting in a high and stable yield. It is worth noting that the use of magnetized oxygenated water for irrigating rice and wheat in this study did not significantly promote their growth and even had some inhibitory effect, which might be related to the change in the properties of the water.

### 4.2. Effect of Activated Water on Root Characteristics of Wheat and Rice

Roots are essential tissue organs for plants to obtain water, nutrients, and oxygen from the soil, and they are also essential synthesis sites for some plant hormones [53]. Its morphological and physiological characteristics strongly correlate with the growth and development, yield, and quality of the aboveground parts of plants [54]. In this study, we found that aerated water significantly promoted the growth of rice roots, and magnetized water significantly promoted the growth of wheat roots. This may be explained by the inherent characteristics of rice and wheat; for example, the root distribution of rice is shallow, with more than 85% of the roots distributed in the 0–20 cm soil layer [55], while wheat roots penetrate deeper into the soil layer, up to 200 cm deep [56]. Furthermore, rice is a semi-aquatic plant and requires sufficient oxygen during growth. Suppose the oxygen supply to the root zone is insufficient. In that case, it will promote ethylene synthesis and lead to the formation of root aeration tissue, limiting root growth and directly affecting the plant's growth, yield, and quality [57]. Finally, the highest oxygen uptake by the rice root system is from the booting stage to the heading stage, which is also the peak water demand period in the rice's life [58]. The traditional method of increasing oxygen to the root zone by draining and exposing the field cannot resolve this conflict, and aerated water irrigation just effectively avoids it. Zhu et al. [19] found that aeration treatment significantly increased soil oxygen content and soil oxidation-reduction potential in the root zone, especially for the heading stage. The increase in soil oxygen content is conducive to the emergence and growth of roots and the maintenance of high root activity [19]. We also found that irrigation with aerated water improved root traits, including increased root dry biomass, maximum root length, total root length, total root surface area, total root volume, and the proportion of fine roots (D $\leq$ 0.5 mm). Fine roots, known as active roots, are the primary tissues by which plants absorb water and nutrients from soil [55]. Root surface area reflects root absorption capacity, so root diameter and root surface area indicate root activity, which increases with the increase in root surface area [58]. The improvement of root characteristics by aerated water irrigation may be due to the increase in the permeability of the root surface and fine roots, thus forming a morphological structure conducive to the absorption of water and nutrients and was beneficial for rice root dry matter accumulation [58]. Additionally, although magnetized water promoted the root growth of rice seedlings, the difference was not significant. A possible explanation for this phenomenon is that the significant effect of magnetized water on rice occurred mainly after the tillering stage, but our study ended before the tillering stage of rice (Table 1). This is because the study by Zhu et al. [59] showed that the increase in rice yield by magnetized water irrigation was mainly due to the fact that magnetized water increased the number of tillers and percentage of earbearing tiller of rice, and increased the chlorophyll content of rice leaves at booting stage, thus increasing the rice set rate. At the same time, they also found that magnetized water had no significant effect on the dry matter production of rice at the tillering stage. It has been shown in this study that magnetized aerated water irrigation reduced rice roots' growth and biomass formation compared with magnetized water irrigation and aerated water irrigation, indicating that magnetized aerated water could not superimpose the effects of magnetized water and aerated water. The possible reason might be that the effects of magnetization or aeration are counteracted during the process of magnetization and then aeration of the water. In the future, the properties of magnetized aerated water such as the

surface tension, viscosity coefficient, dissolved oxygen concentration, and other indicators need to be measured to explain this phenomenon specifically.

On the other hand, wheat is a terrestrial plant and consumes less oxygen than rice [56]. Therefore, the effect of aerated water on the root growth of wheat is limited. Lu et al. [35] treated two wheat varieties with aeration by adding different concentrations of $H_2O_2$ and found that the root lengths of Ningchun and Xihan seedlings were significantly reduced under $H_2O_2$ treatment, with Ningchun being more sensitive to $H_2O_2$. This showed that the effect of aeration treatment is related to the method of aeration and crop species. Our results show that magnetized water irrigation significantly increased the maximum root length, total root volume, and the proportion of fine roots ($D \leq 0.5$ mm) in wheat. These results are consistent with Zhao et al. [21] that irrigation with magnetized water promotes wheat root development, including root length, root surface area, and root tips. Similarly, Anjali Anand et al. [60] used a magnetic field of 100–200 mT to magnetize maize seeds. They also found that the magnetization treatment significantly promoted the growth of maize roots in terms of length of longest root, total root length, root surface area, mean root diameter, and root-shoot ratio. Liu et al. [61] found that the root length, surface area, mean diameter, number of tips, and short roots number of *Populus euramerican* root irrigated with magnetized water significantly increased by 32.9–40.4%, 37.4–41.7%, 4.1–12.7%, 11.9–26.5% and 11.9–41.6%, respectively, compared with non-magnetized water treatments. The promoting effect of magnetized water on the wheat root system can be explained as follows: First, the physicochemical properties of water are changed by magnetization, which promotes the absorption and utilization of water by the root system [18,62]. Second, magnetized water irrigation increased the soil water content, promoted the formation of soil aggregates, thereby creating a soil environment conducive to root growth [21,63,64]. Finally, the magnetized water irrigation increased the uptake of the $Ca^{2+}$ ions in rice seedlings and resulted in better growth of meristematic tissues in the root [59]. In addition, our study found that the magnetized aerated water treatment reduced all indices of wheat roots compared to other treatments and inhibited the growth of wheat roots, consistent with the results for rice. Since this study was conducted under hydroponic conditions, only the seedling growth of two crops was considered and the effect on crop yield was not considered. Therefore, the results of this study can only provide a partial reference for the application of activated water under hydroponics. We will repeat similar experiments using soil in the future to compare or validate the results of this study and to focus on the effect of activated water on crop yield. Furthermore, future research is warranted to investigate the effects of activated water in production practice from large-scale use in the field and strengthen research on the mechanism of aerated water under hypoxic stress conditions.

## 5. Conclusions

Different types of activated water (magnetized water, aerated water, and magnetized aerated water) had different effects on the growth of rice and wheat seedlings in hydroponics. Compared with the control, aerated water and magnetized water irrigation significantly increased the leaf area and biomass of rice seedlings, and magnetized aerated water irrigation promoted the growth of rice seedlings. Still, aerated water and magnetized aerated water reduced each growth parameter of wheat seedlings, and magnetized water did not significantly promote the growth of wheat seedlings. The biomass, maximum root length, total root length, and total root surface area of rice seedlings under the irrigation treatment with aerated water significantly increased by 22.4%, 15.4%, 33.8%, and 27.8%, respectively. Aerated water mainly enhances the growth of rice seedlings under hydroponic conditions by promoting root growth, especially the growth of fine roots ($D \leq 0.5$ mm). Magnetized water and magnetized aerated water also stimulated the roots growth of rice, but the differences were not statistically significant. The magnetized water irrigation promoted the root system of wheat seedlings. In contrast, the aerated water and magnetized aerated water had an inhibitory effect on the growth of wheat roots in hydroponics. Further research is

needed to clarify the effect of activated water on different crops to provide a reference for the application of activated water in agricultural production.

**Author Contributions:** Conceptualization, X.Y. and J.F.; methodology, X.Y. and J.F.; software, X.Y.; formal analysis, X.Y.; data curation, X.Y.; investigation, X.Y. and J.G.; writing—original draft preparation, X.Y., J.G. and Z.L.; writing—review and editing, X.Y. and J.F.; visualization, X.Y.; supervision, J.F.; project administration, J.F.; funding acquisition, J.F. All authors have read and agreed to the published version of the manuscript.

**Funding:** This research was funded by the National Natural Science Foundation of China (41830754).

**Institutional Review Board Statement:** Not applicable.

**Informed Consent Statement:** Not applicable.

**Data Availability Statement:** Not applicable.

**Conflicts of Interest:** The authors declare no conflict of interest.

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
