# Peer review of "Effect of Irrigation with Activated Water on Root Morphology of Hydroponic Rice and Wheat Seedlings"

_agronomy, doi:10.3390/agronomy12051068_

Round 1

Reviewer 1 Report

Greeting Authors,

Thanks for the great work. I have the following comments.

1- you have not started the sentence with (We or it) because it makes it a weak sentence. 

2- How can we use this method in the field? What do you think is the result?

3- What do you thank the crop yield increase or decreases? you take just green growth in this experiment. 

4- The discussion part needs more deep discussion information about why you have a significant result or not? Also, you used the word (We guess the possible reason is that the effects of magnetization or aeration are counteracted during the process of magnetization and then aeration of the water.) the Guess word is not a scientific discussion.

What do you think is going to happen if we apply this experiment in the field? Do you think the soil properties (physical, chemical, and Biological) affect the result or not? Because you have to do another experiment with soil to compare to hydroponic experimental. 

Author Response

Dear reviewer,

We feel immense gratitude for your previous review work on our manuscript, “Effect of irrigation with activated water on root morphology of hydroponic rice and wheat seedlings” (agronomy-1641732). We considered all of the comments and suggestions carefully and incorporated necessary changes in the revision.

A summary of how we addressed your suggestions, point by point, is included in this letter. Attached is a marked manuscript with major corrections highlighted in red.  We are grateful that you will reconsider our revised manuscript.

With thanks,

Authors

Reviewer:

Thanks for the great work. I have the following comments.

1- you have not started the sentence with (We or it) because it makes it a weak sentence.

Authors’ response: Yes, we agreed. We have modified some sentences so that they start with we or it. “We found that plant height, leaf area, and aboveground biomass of rice…” See line 283. “It also showed that the magnetized water and magnetized aerated water promoted the growth…” See line 320-321. “We also found that irrigation with aerated water…” See line 389-390. “It has been shown in this study that magnetized aerated water irrigation reduced rice roots’ growth and biomass formation compared with magnetized water irrigation and aerated water irrigation,…”See line 411-414.

2- How can we use this method in the field? What do you think is the result?

Authors’ response: In the Discussion (4.1.) with references to the relevant literature, we add a general description of their approach in using magnetized or aerated water applied in the field and show the results they obtained to show how this approach is used in the field and what the results are. As follows.

 “Several studies have applied aerated water by micro and nano bubble generators to greenhouse lettuce [15], tomato [40], and cucumber [41], and others have obtained aerated water by combining subsurface drip irrigation systems with air injection systems to irrigate field corn [42]. All these results show that aerated water irrigation can promote crop growth and improve crop yield.” See line 294-298.

 “In this case, the magnetized water was usually obtained in the field by placing a permanent magnet on the irrigation water pipe.” See line 355-356.

3- What do you think the crop yield increase or decreases? you take just green growth in this experiment.

Authors’ response: In this study, we only considered the effect of activated water on seedling growth of rice and wheat without considering crop yield, which is one of our weaknesses. We have shown this information in Discussion (4.2.) and have presented some ideas for our future research. “Since this study was conducted under hydroponic conditions, only the seedling growth of two crops was considered and the effect on crop yield was not considered.” See line 446-448. “We will repeat similar experiments using soil in the future to compare or validate the results of this study and to focus on the effect of activated water on crop yield.” See line 449-451.

4- The discussion part needs more deep discussion information about why you have a significant result or not? Also, you used the word (We guess the possible reason is that the effects of magnetization or aeration are counteracted during the process of magnetization and then aeration of the water.) the Guess word is not a scientific discussion.

Authors’ response: Yes, we agreed. We have added some information in the Discussion (4.2.) to explain why magnetized water has no significant effect on rice. “Additionally, although magnetized water promoted the root growth of rice seedlings, the difference was not significant. A possible explanation for this phenomenon is that the significant effect of magnetized water on rice occurred mainly after the tillering stage, but our study ended before the tillering stage of rice (Table 1). This is because the study by Zhu et al. [59] showed that the increase in rice yield by magnetized water irrigation was mainly due to the fact that magnetized water increased the number of tillers and percentage of earbearing tiller of rice, and increased the chlorophyll content of rice leaves at booting stage, thus increasing the rice set rate. At the same time, they also found that magnetized water had no significant effect on the dry matter production of rice at the tillering stage.” See line 402-411.

Also, we have deleted “guess” and rewritten the sentence as “The possible reason might be that…” See line 415.

What do you think is going to happen if we apply this experiment in the field? Do you think the soil properties (physical, chemical, and Biological) affect the result or not? Because you have to do another experiment with soil to compare to hydroponic experimental.

Authors’ response: Yes, we agreed. We will conduct corresponding experiments in soil to compare with each other, and we show this idea in the Discussion (4.2.). “We will repeat similar experiments using soil in the future to compare or validate the results of this study and to focus on the effect of activated water on crop yield.” See line 449-451.

Reviewer 2 Report

Yang et al., in the paper titled "Effect of irrigation with activated water on root morphology of hydroponic rice and wheat seedlings," studied the effect of magnetized water, aerated water, and magnetized water treated hydroponically on the growth of rice and wheat seedlings and root growth.

This study shows different activated water irrigation shows the different effects on rice and wheat seedling growth. While the study describes the phenotypic characterization well, a lot of aspects are still not clearly explained in this study. In this form, the novelty of this study is still not very clear and inclined to common descriptive studies. 

Here is detailed list of my comment

Major:

  1. Arguably there are still questionable approaches and results in this study. For instance,  in the abstract authors expected to investigate the effect of treatment and explore the mechanism affecting crop growth. However, author did not provide enough clear mechanism of the process and main component affecting the activated water mechanism.
  2. Author did not characterize the treatment parameter, such as the physicochemical characteristics of the activated water (e.g. O2 saturation, water surface tension, etc.) this data is important to be able to explain the mechanism underlying the treatment effect. 
  3. While the author provides the possible mechanism through citation of several papers, the paper cited also not clearly explained the mechanism that the author intended to use as the bases of the argument. 
  4. While the authors show both promote effect and inhibitory effect, the discussion of why this phenomenon occurs are still can be explained more. 
  5. Is there any supporting reference to back up your guess at line 371-372. It is seems will be very usefull if you could also discuss more the parameters that clearly mentioned in the previous findings. 
  6. SInce the title is root morphology, I was expecting the author could show the morphological differences clearly in the roots. then show the quantification. 
  7. Is there any long-term effect that you could see in the experiment, such as other studies have been shown yield effect or grain effect. 
  8. What is the term fine root? is it root hair? 
  9. In the method, how many individual plants were used for the experiment? It is better to clearly state the replication. For example, for root morphology data the sample was collected from xxxx part of the root from each plant; quantification was performed xxx times in one plant, and the mean from one plant was then counted as one biological replicate. total biological replicates xxxx for the graph. 
  10. Statistical significance is missing in Fig. 5. 

Author Response

Dear reviewer,

We feel immense gratitude for your previous review work on our manuscript, “Effect of irrigation with activated water on root morphology of hydroponic rice and wheat seedlings” (agronomy-1641732). We considered all of the comments and suggestions carefully and incorporated necessary changes in the revision.

A summary of how we addressed your suggestions, point by point, is included in this letter. Attached is a marked manuscript with major corrections highlighted in red.  We are grateful that you will reconsider our revised manuscript.

With thanks,

Authors

Reviewer:

Yang et al., in the paper titled "Effect of irrigation with activated water on root morphology of hydroponic rice and wheat seedlings," studied the effect of magnetized water, aerated water, and magnetized water treated hydroponically on the growth of rice and wheat seedlings and root growth.

This study shows different activated water irrigation shows the different effects on rice and wheat seedling growth. While the study describes the phenotypic characterization well, a lot of aspects are still not clearly explained in this study. In this form, the novelty of this study is still not very clear and inclined to common descriptive studies.

Here is detailed list of my comment

Major:

1. Arguably there are still questionable approaches and results in this study. For instance, in the abstract authors expected to investigate the effect of treatment and explore the mechanism affecting crop growth. However, author did not provide enough clear mechanism of the process and main component affecting the activated water mechanism.

Authors’ response: Yes, we agreed. We have deleted “and explore the mechanisms of activated water affecting crop growth” from the Abstract.

In addition, we deleted “analyze the effect mechanism of activated water on seedling growth” in the Introduction, and instead with “investigate the effect of different types of activated water on the root characteristics of rice and wheat in an attempt to explain why activated water affects seedling growth.” See line 102-104.

2. Author did not characterize the treatment parameter, such as the physicochemical characteristics of the activated water (e.g. O2 saturation, water surface tension, etc.) this data is important to be able to explain the mechanism underlying the treatment effect.

Authors’ response: We have explained this issue in the Materials and Methods. “The principles of MW and AW production are the same as those of zhao et al. [21] and zhu et al. [19] respectively, so their conclusions regarding the physicochemical properties of water are applicable to the present study.” See line 140-143.

3. While the author provides the possible mechanism through citation of several papers, the paper cited also not clearly explained the mechanism that the author intended to use as the bases of the argument.

Authors’ response: Yes, we agreed. First, we have added two references, which we hope will be more convincing. See line 440, 442.

21.  Zhao, G.Q.; Mu, Y.; Wang, Y.H.; Wang, L. Magnetization and Oxidation of Irrigation Water to Improve Winter Wheat (Trit-icum Aestivum L.) Production and Water-Use Efficiency. Agricultural Water Management 2022, 259, 107254, doi:10.1016/j.agwat.2021.107254.

62.  Surendran, U.; Sandeep, O.; Joseph, E.J. The Impacts of Magnetic Treatment of Irrigation Water on Plant, Water and Soil Characteristics. Agricultural Water Management 2016, 178, 21–29, doi:10.1016/j.agwat.2016.08.016.

Second, we have added the following description in the Introduction. “The mechanism of crop growth affected by activated water is not fully clear.” See line 93-94.

4. While the authors show both promote effect and inhibitory effect, the discussion of why this phenomenon occurs are still can be explained more.

Authors’ response: Yes, we agreed. We have added some information in the Discussion (4.2.) to explain why magnetized water has no significant effect on rice. “Additionally, although magnetized water promoted the root growth of rice seedlings, the difference was not significant. A possible explanation for this phenomenon is that the significant effect of magnetized water on rice occurred mainly after the tillering stage, but our study ended before the tillering stage of rice (Table 1). This is because the study by Zhu et al. [59] showed that the increase in rice yield by magnetized water irrigation was mainly due to the fact that magnetized water increased the number of tillers and percentage of earbearing tiller of rice, and increased the chlorophyll content of rice leaves at booting stage, thus increasing the rice set rate. At the same time, they also found that magnetized water had no significant effect on the dry matter production of rice at the tillering stage.” See line 402-411.

5. Is there any supporting reference to back up your guess at line 371-372. It is seems will be very usefull if you could also discuss more the parameters that clearly mentioned in the previous findings.

Authors’ response: We unfortunately could not find relevant literature to support my speculation because there are few studies on magnetized aerated water (water is first magnetized and then aerated). Moreover, we have added the relevant information in the Introduction. “Moreover, we found few studies combining two different types of activated water to study their superimposed effects.” See line 92-93.

Furthermore, we have added a detailed description of the changes in the properties of activated water in the Introduction. As follows.

“Otsuka and Ozeki [20] found that the contact angle of distilled water was significantly reduced from 65° to 57.5° after magnetization. The results of Esmaeilnezhad et al. [18] showed that the surface tension of water decreased and the pH increased after magnetization of both pure and saline water. Zhao et al. [21] concluded that magnetization and oxidation of water increased pH and maintained it for about 60 h, reduced surface tension and viscosity coefficient by 7.4–15.4%, maintained it for about 8–10 h, increased dissolved oxygen concentration, and detected the formation of hydroxyl radicals.” See line 57-63.

6. SInce the title is root morphology, I was expecting the author could show the morphological differences clearly in the roots. then show the quantification.

Authors’ response: Yes, we agreed. We have added or rewritten some sentences in Results (3.3.2. and 3.3.3.) to show the morphological differences of the roots more clearly, as shown below.

“The total root length and total root surface area under AW irrigation were 2870 cm and 177 cm2, respectively, which were significantly increased by 33.8% and 27.8% compared with CK (2145 cm and 139 cm2).” See line 238-240.

 “While the total root length and total root surface area under MW and MAW irrigation were 2266 cm, 2346 cm and 155 cm2, 153 cm2, respectively, which increased by 5.7–11.9% than CK, with no significant difference between treatments.” See line 240–243.

 “The percentage of roots with diameters of 0.0–0.5 mm and 0.5–1.0 mm in AW was 93.0% and 6.8%, respectively, showing significant differences from CK.” See line 269-271.

7. Is there any long-term effect that you could see in the experiment, such as other studies have been shown yield effect or grain effect.

Authors’ response: Yes. We have added or rewritten some sentences in Discussion (4.1.) to show the long-term effects of activated water applications in the field, as follows.

“Moreover, Pendergast et al. [43] applied aerated water to drip irrigation on broadacre cotton for seven growing seasons (2005–06 to 2012–13), and found that the seven-year average yield was significantly higher by 10% compared to the control. Dhungel et al. [44] used Mazzei air injectors to obtain aerated water and irrigated field pineapples for 5 consecutive years (2007–2011). They arrived at a total industrial fruit yield of 73.3 t ha−1 for the aerated treatment, which was 6% higher than the control.” See line 298-303.

“The results were consistent with those reported by Zhao et al. [49], who conducted a two-year (2018–19 to 2019–20) field experiments and found that irrigation with magnetized water can significantly promote winter-wheat growth, increasing yield by 10.0% and 11.1%, and improve water use efficiency and irrigation water use efficiency.” See line 348-352.

“Meanwhile, Hozayn et al. [50] also conducted a two-year (2009–10 to 2010–11) field trial using canola, and found that the seed yield and oil yield under magnetized water irrigation were 697.0 kg fed−1 and 223.0 kg fed−1, respectively, which increased by 38.7% and 58.5% compared to normal water.” See line 352-355.

8. What is the term fine root? is it root hair?

Authors’ response: We are sorry that the statement was unclear. In this study, we considered root systems with diameters less than 0.5 mm as fine roots, which include root hairs. In order to express this term clearly, we have qualified all words containing fine roots in the article by adding “(D≤0.5 mm)” after them. See line 22, 274, 392, and 466-467.

9. In the method, how many individual plants were used for the experiment? It is better to clearly state the replication. For example, for root morphology data the sample was collected from xxxx part of the root from each plant; quantification was performed xxx times in one plant, and the mean from one plant was then counted as one biological replicate. total biological replicates xxxx for the graph.

Authors’ response: Yes, we agreed. To clearly state the trial replicates, first, we added “, i.e., each treatment included 16 plants” to the Materials and Methods (2.2.) to explain how many plants were used in the experiment, see line 126.

Second, we added the following information to Materials and Methods (2.4.) to clearly state the biological replicates of each data.

“The plant height and the longest root length of each plant were measured three times and the mean from one plant was then counted as one biological replicate. Total biological replicates 16 for the graph.” See line 147-149.

“These data were measured once per plant, and the average of 16 plants was used for the graph.” See line 153-154.

“The roots of each plant were scanned once, and 8 plants were randomly selected for scanning under one treatment, so the results obtained were the average of the 8 plants.” See line 159-161.

“The mean of a total of 16 plants per treatment was taken for graphing.” See line 162-163.

Finally, we revised “Roots” in line 154-155 to “All roots of one plant” and “dry weight” in line 162 to “aboveground and root dry weights of each plant” to make it clearer.

10. Statistical significance is missing in Fig. 5.

Authors’ response: Thanks for your reminder. We have added statistical significance to Fig. 5. See line 277.

Reviewer 3 Report

The authors should put their new investigation in the context of their earlier similar studies. What are the new phenomena they expect to clarify.

In the introduction the authors list couple of properties of the treated water, however in the work there are no data given about the treated water used in the investigations. Without these data the studies are lacking grounds.

On Fig 5 data are missing.

Author Response

Dear reviewer,

We feel immense gratitude for your previous review work on our manuscript, “Effect of irrigation with activated water on root morphology of hydroponic rice and wheat seedlings” (agronomy-1641732). We considered all of the comments and suggestions carefully and incorporated necessary changes in the revision.

A summary of how we addressed your suggestions, point by point, is included in this letter. Attached is a marked manuscript with major corrections highlighted in red.  We are grateful that you will reconsider our revised manuscript.

With thanks,

Authors

Reviewer:

The authors should put their new investigation in the context of their earlier similar studies. What are the new phenomena they expect to clarify.

Authors’ response: Yes, we agreed. We have added or rewritten some sentences in the Introduction to show the new phenomena of our study. As follows.

“Moreover, we found few studies combining two different types of activated water to study their superimposed effects.” See line 92-93.

“In this study, we used hydroponics to investigate the effects of magnetized and aerated water on the growth of rice (Oryza sativa L.) and wheat (Triticum aestivum L.), and also explored whether combining magnetized water with aerated water (magnetized aerated water) showed a superimposed effect on crop growth.” See line 97-100.

In the introduction the authors list couple of properties of the treated water, however in the work there are no data given about the treated water used in the investigations. Without these data the studies are lacking grounds.

Authors’ response: We have explained this issue in the Materials and Methods. “The principles of MW and AW production are the same as those of zhao et al. [21] and zhu et al. [19] respectively, so their conclusions regarding the physicochemical properties of water are applicable to the present study.” See line 140-143.

On Fig 5 data are missing.

Authors’ response: We are so sorry for this mistake. We have modified the image file format to avoid the error from happening again. See line 277.